# HUMANOID-R0: BRIDGING TEXT-TO-MOTION GENERATION AND PHYSICAL DEPLOYMENT VIA RL

## ABSTRACT

Motion generation models aim to create diverse and realistic motions from inputs such as text or keyframes, with applications in gaming, animation, and 3D content creation. Recently, generated motions have emerged as a viable supplement to reinforcement learning and teleoperation for humanoid control, overcoming the limitations of simulation diversity and bypassing the need for expensive demonstration collection. However, these downstream applications impose stringent requirements on motion quality—specifically, smoothness, skeletal plausibility, and keyframe consistency. To ensure generated motions meet these criteria, we quantify these metrics and convert them into reward functions for RL fine-tuning. Specifically, we fine-tune the motion generation component of HumanoidVLA using GRPO, resulting in Humanoid-R0, a model whose outputs are well-suited for robot control. Our approach is rigorously validated through extensive metric evaluations, simulation rendering, and real-world deployment, demonstrating significant improvements in motion smoothness, plausibility, and consistency. Notably, Humanoid-R0 enables stable execution of challenging sequences of consecutive commands on a real G1 robot, showcasing its enhanced capability for long-horizon task completion.

## 1 INTRODUCTION

Motion generation models have revolutionized digital content creation by synthesizing diverse and realistic motions from textual descriptions or sparse keyframes, enabling breakthroughs in gaming, animation, and 3D virtual environments (Arikan & Forsyth, 2002; Min et al., 2009). These models excel at capturing stylistic variations and natural movement patterns, yet their potential extends far beyond purely visual applications. Recent advancements in robotics have begun to take advantage of generative motion paradigms to achieve flexible humanoid control as a novel alternative to traditional strategies such as reinforcement learning (RL) and teleoperation (Zhang et al., 2024; Mao et al., 2024). While RL-based methods often struggle with limited simulation-environment diversity and complex reward function designs (Hester et al., 2010), and teleoperation relies on expensive resource-intensive demonstration capture (Darvish et al., 2023), generative approaches offer a promising pathway to scalable, data-efficient motion synthesis for flexible humanoid behavior control.

However, deploying generative motions for physical humanoid robots introduces stringent quality requirements that transcend conventional standards. In virtual domains, kinematic discontinuous or even abrupt motion, as well as abnormal skeletal proportions, will not lead to catastrophic consequences and may even remain visually imperceptible. But in physical humanoid robot, such artifacts can induce cascading failures and even cause permanent actuator damage. Moreover, when generating motions for controlling robots, it is crucial to consider the current posture of the robot. That is, the first few frames of the generated motion should be very close to the current posture. We systematically formalize these requirements as three critical metrics:

- **Smoothness** is a crucial aspect in robot control, requiring that the magnitude of motion changes between consecutive actions should not be too large; otherwise, it can easily damage the joints.

- **Skeletal plausibility** determines whether the motions can match the robot's physical structure. For instance, overly short joint lengths may cause the robot to meaninglessly bend its limbs.

- **Keyframe consistency** demands that the initial frame of the generated motion aligns with the robot's current posture. Since abrupt motion changes on a real robot can lead to incalculable losses.

Current motion generation models, including SOTA like HumanoidVLA (Ding et al., 2025) and UH-1 (Mao et al., 2024) designed for humanoid robot control, all do not explicitly account for the additional requirements that robots impose on motions. To bridge this gap, we first quantify the three aforementioned metrics and then convert them into reward functions for fine-tuning via RL. Moreover, considering that collecting large-scale, high-quality motion data that is both human-like and robot-feasible is prohibitively expensive and time-consuming, we innovatively transform the supervised fine-tuning (SFT) loss into an SFT reward and incorporate it into the RL fine-tuning phase. This allows us to treat the existing human motion data as a weak prior, guiding semantic alignment while using RL to enforce critical physical constraints in a single training stage. Specifically, we select the motion generation component of HumanoidVLA as our base model and RL fine-tune it using Group Relative Policy Optimization (GRPO) (Shao et al., 2024) to obtain our model **Humanoid-R0**.

We evaluate Humanoid-R0 on classic datasets using traditional metrics, including the FID score for motion quality, the MMDist metric for motion-text alignment, and metrics relevant to humanoid robot control including smoothness and skeleton plausibility. Humanoid-R0 achieve SOTA performance across these metrics. Experiments in a simulated environment further demonstrate that Humanoid-R0, incorporating the keyframe consistency reward, can effectively consider the robot's current posture, avoiding falls when continuously executing commands. Finally, we deploy our Humanoid-R0 on the G1 robot, which is tasked with executing a challenging long-horizon instruction-following task involving 7 consecutive commands. The generated motions from Humanoid-R0 are executed on the physical robot through a low-level PD controller. Under this pipeline, the robot smoothly executes the commands and their transitions. Please refer to our video in the supplementary materials.

## 2 RELATED WORKS

### 2.1 HUMAN MOTION GENERATION

Human motion generation has evolved from kinematic modeling to physics-aware synthesis. Early probabilistic models (Hidden Markov Models (Brand & Hertzmann, 2000), Gaussian Processes (Wang et al., 2007)) struggle with high-dimensional sequences, while RNNs introduce temporal modeling for locomotion (Holden et al., 2017). Transformer-based models, such as ACTOR (Petrovich et al., 2021), MotionBERT (Zhu et al., 2023), MDM (Tevet et al., 2022) and hierarchical architectures(Xu et al., 2020; Bie et al., 2022), further improve global coherence. Diffusion models (MotionDiffuse (Zhang et al., 2024), ReMoDiffuse (Zhang et al., 2023b), B2A-HDM (Xie et al., 2024)) enable high-fidelity text-to-motion generation. However, most methods prioritize perceptual quality over physical plausibility. Recent physics-aware works like PhySHOI (Wang et al., 2023), PADL (Juravsky et al., 2022), PhysDiff (Yuan et al., 2023) enforce biomechanical constraints via adversarial training or reinforcement learning. These methods primarily synthesize motions for virtual humans, emphasizing visual fidelity rather than robotic deployment constraints, leaving a critical sim-to-real gap.

### 2.2 HUMAN MOTION FOR HUMANOID CONTROL

Traditional model-based methods for humanoid control ensure stability but require precise dynamics models and struggle to generalize due to scarce humanoid data (Li et al., 2023a; Kuindersma et al., 2016; Elobaid et al., 2023; Dantec et al., 2021; Dai et al., 2014). Retargeting abundant human motions to humanoid robots has emerged as a promising paradigm. Works like Exbody (Cheng et al., 2024; Ji et al., 2024), HARMON (Jiang et al., 2024), PHC (Luo et al., 2023), and H2O (He et al., 2024b;a) map human motions to robots via SMPL (Loper et al., 2023), enabling flexible control. However, these methods often neglect real-world physical constraints, leading to unstable or infeasible motions when deployed on physical robots (Chen et al., 2024; Li et al., 2023b). Even language-guided approaches (He et al., 2024a; Jiang et al., 2024; Mao et al., 2024) generate motions passively without considering embodied dynamics. To address this, we integrate physical constraints via RL fine-tuning, which ensures motions adhere to hardware limits while preserving human-like semantics.

## 3 FRAMEWORK

Our framework builds on a text-motion pre-aligned large motion model and aims to bridge the gap between language-driven motion generation and real-world humanoid control. We first introduce an

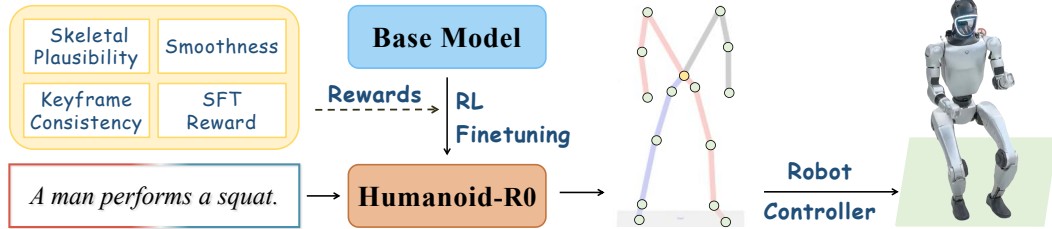

Figure 1: Overall Framework. The Humanoid-R0 module refers specifically to the fine-tuned motion generator. Our method refines a text-to-motion generator for real-world deployment using physics-aware rewards. It follows a two-stage pipeline: (1) language-conditioned keypoint generation by the motion model; (2) goal-conditioned policy adaptation via a low-level PD controller.

instruction-to-action pipeline that maps natural language commands to robot-executable actions. We then define three motion metrics related to humanoid robot control, which can also serve as reward functions for reinforcement learning (RL) fine-tuning. Finally, We describe the detailed process of RL fine-tuning the motion model and explain why we innovatively integrate the supervised fine-tuning (SFT) phase into the RL fine-tuning. The overall framework can be seen in Figure 1.

## 3.1 INSTRUCTION-TO-ACTION PIPELINE FOR HUMANOID CONTROL

The task of instruction-based humanoid control defines a structured framework that translates natural language commands into robot-executable actions. The process follows a two-stage pipeline:

$$\pi_{\text{Text2Motion}} : \mathcal{T} \mapsto \hat{\mathcal{P}}_{robot}, \quad \pi_{\text{Motion2Action}} : \hat{\mathcal{P}}_{robot} \mapsto \mathcal{A}_{robot}, \tag{1}$$

where $\mathcal{T}$ denotes the input text command, $\hat{\mathcal{P}}_{robot}$ represents the generated keypoint trajectory, and $\mathcal{A}_{robot}$ specifies the low-level target DoF positions for direct execution on the physical robot.

**(1) Base Motion Generator** In the first stage, a pre-aligned large model serves as the foundation for mapping natural language instructions to high-quality motion trajectories. This model leverages large-scale video data with language annotations to establish an initial alignment between text and motion. To efficiently represent human poses, it adopts a decompositional compression approach by dividing the body into five functional regions: left leg, right leg, torso, left arm, and right arm, each encoded separately using a VQ-VAE architecture. Specifically, the encoder compresses a pose sequence $\mathcal{P}$ into continuous latent features $\mathcal{Z} = F_{\text{encoder}}(\mathcal{P}) = \{\mathcal{Z}^b\}_{b=1}^5$, which are then quantized to produce discrete token representations $\mathcal{Z}' = F_{\text{quant}}(\mathcal{Z})$. The decoder reconstructs the original motion from these tokens as $\hat{\mathcal{P}} = F_{\text{decoder}}(\mathcal{Z}')$.

The generated motion is represented as $\hat{\mathcal{P}}_t \in \mathbb{R}^{T \times (15 \times 3)}$, where $T$ denotes the number of time steps and each joint has $(x, y, z)$ coordinates (see Figure 2 for details). While many existing motion generation models use 24 joints to capture richer motion details (Jiang et al., 2023; Zhang et al., 2024), we find that many of these joints are relatively redundant due to the limitations of the degrees of freedom of the real humanoid robot. After weighing the trade-off between simplicity and expressiveness of motion representation, we select 15 joints. The VQ-VAE is trained using a combination of reconstruction loss $\mathcal{L}_{\text{rec}}$, embedding loss $\mathcal{L}_{\text{emb}}$, and commitment loss $\mathcal{L}_{\text{com}}$, formulated as:

$$\mathcal{L}_{\text{vqvae}} = \underbrace{\|\mathcal{P} - \hat{\mathcal{P}}\|_2}_{\mathcal{L}_{\text{rec}}} + \underbrace{\|\text{sg}(\mathcal{Z}) - \mathcal{Z}'\|_2}_{\mathcal{L}_{\text{emb}}} + \underbrace{\|\mathcal{Z} - \text{sg}(\mathcal{Z}')\|_2}_{\mathcal{L}_{\text{com}}}, \tag{2}$$

Figure 2: The Motion contains 15 joints.

where $\text{sg}(\cdot)$ indicates stop-gradient operation, ensuring clear separation of training objective.

Conditioned on text commands $\mathcal{T}$, the large model autoregressively generates sequences of quantized keypoint tokens $\mathcal{Z}' = \{z_1', z_2', \ldots, z_N'\}$, modeling the conditional probability $P(z_i'|z_{1:i-1}', l)$, where $z_i'$ is the current token to be predicted, $z_{1:i-1}'$ provides historical context, and $l$ is the text embedding. The learning objective minimizes the negative log-likelihood:

$$\mathcal{L}_{\text{LLM}} = -\sum_i \log P(z_i'|z_{1:i-1}', l). \tag{3}$$

The final motion trajectory $\hat{\mathcal{P}}_{robot}$ is reconstructed from $\mathcal{Z}'$ using the VQ-VAE decoder.

**(2) Humanoid Robot Controller** While the generated keypoints $\hat{\mathcal{P}}_{robot}$ capture semantically meaningful motions, they cannot be directly deployed on a physical robot due to limitations in physical realizability and safety guarantees. Therefore, we use a goal-conditioned control policy $\pi_{\text{Motion2Action}}$ to adapt the motions for reliable execution:

$$\pi_{\text{Motion2Action}} : \mathcal{G} \times \mathcal{O} \mapsto \mathcal{A}_{robot}, \tag{4}$$

where the goal space $\mathcal{G}$ includes the reference keypoint trajectory $\hat{\mathcal{P}}_{robot}$ and root movement goals, and the observation space $\mathcal{O}$ contains proprioceptive information such as root orientation and joint states. The output $\mathcal{A}_{robot}$ defines target DoF positions, which are transformed into motor torques via PD controller. We train this policy using Proximal Policy Optimization (PPO), ensuring that the resulting actions $\mathcal{A}_{robot}$ not only align with the desired motion but also satisfy physical constraints such as stability and contact feasibility. This unified pipeline enables our system to generate motions that are both semantically consistent with the input text and physically deployable on real robots.

### 3.2 Physics-Aware Rewards for Deployable Motion

**a) Smoothness.** To ensure natural and seamless transitions between consecutive timesteps, we design a smooth reward term that penalizes abrupt variations in the generated motion. This is achieved by evaluating the discrepancy between keypoints at adjacent frames. Formally, the smooth reward is:

$$\mathcal{R}_{\text{smooth}} = -\frac{1}{T-1} \sum_{t=1}^{T-1} \|\hat{\mathcal{P}}_{t+1} - \hat{\mathcal{P}}_t\|_2, \tag{5}$$

where $\hat{\mathcal{P}}_t$ and $\hat{\mathcal{P}}_{t+1}$ represent the predicted keypoints at timesteps $t$ and $t+1$, respectively. The model encourages smaller differences between adjacent frames, resulting in higher rewards for smoother motions. This promotes fluid and natural motion transitions, making the generated motions easier to deploy on physical robots while avoiding abrupt changes.

The standard smooth reward may encourage trivial, motionless solutions—a form of reward hacking. To mitigate this, we propose a normalized variant $\mathcal{R}_{\text{smooth}}^+$, defined as:

$$\mathcal{R}_{\text{smooth}}^+ = e^{\left| \frac{\mathcal{R}_{\text{smooth}} - \mu}{\sigma} \right|}, \tag{6}$$

where $\mu$ and $\sigma$ are predefined mean and standard deviation. This ensures the reward has a positive lower bound, effectively preventing static outputs. We compare both variants in our ablation study.

**b) Skeletal Plausibility.** To ensure anatomical consistency between the generated keypoints and the physical robot's structure, we introduce a skeleton reward that penalizes deviations in critical body segment lengths. This reward evaluates six key segments:

| | | |
|---|---|---|
| **Elbow-Hand Left**: Joints 10–11 | **Toe-Ankle Left**: Joints 3-4 | **Knee-Ankle Left**: Joints 2-3 |
| **Elbow-Hand Right**: Joints 13–14 | **Toe-Ankle Right**: Joints 7–8 | **Knee-Ankle Right**: Joints 6-7 |

For each segment, we compute the Euclidean distance between its two endpoints at timestep $t$:

$$d_{i,t} = \|\hat{\mathcal{P}}_t^{(n_1)} - \hat{\mathcal{P}}_t^{(n_2)}\|_2,$$

where $(n_1, n_2)$ denotes the joint indices of the segment (e.g., $(10, 11)$ for Elbow-Hand Left). The reward for each segment is determined by comparing $d_{i,t}$ to a predefined standard length $d_i^*$ using a range-based function:

$$r_{i,t} = \begin{cases} 1 & \text{if } d_{i,t} \in [a_i, b_i], \\ e^{(-k \cdot \min(|d_{i,t} - a_i|, |d_{i,t} - b_i|))} & \text{otherwise}, \end{cases}$$

where $[a_i, b_i]$ defines the valid range around $d_i^*$, and $k$ controls the exponential decay rate for distances outside this range. This design encourages the generated keypoints to match the robot's standardized anatomy. The total skeleton reward is the average of these rewards:

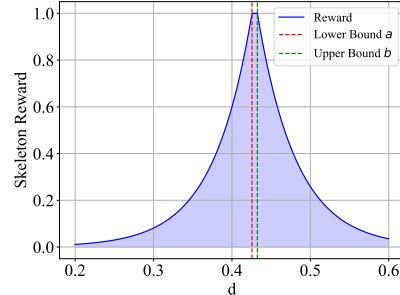

Figure 3: Skeleton reward.

$$\mathcal{R}_{\text{skeleton}} = \frac{1}{6} \sum_{i=1}^{6} r_{i,t}. \tag{7}$$

This reward ensures that the motions are physically plausible and compatible with the robot's structural constraints, facilitating smoother deployment in real-world scenarios.

**c) Keyframe Consistency.** When the model generates motion commands for the robot, it needs to continue generating motions based on the robot's current posture. Otherwise, abrupt changes in motion can make it difficult for the robot to execute the commands, leading to severe consequences. For example, if the robot is currently in an upright stance but the generated motion starts with a squatting posture, the robot is highly likely to fall when attempting to track the squatting posture. We only consider the first four keyframes to determine the consistency reward, formulated as follows:

$$\mathcal{R}_{\text{Consistency}} = \frac{1}{4} \sum_{t=0}^{3} \left\| \mathcal{P}_t^{\text{target}} - \hat{\mathcal{P}}_t \right\|_2, \tag{8}$$

where the target keypoint $\mathcal{P}_t^{\text{target}}$ is determined by the current pose of the robot or motion.

## 3.3 RL FINE-TUNING WITH SFT REWARD

The primary advantage of this unified SFT+RL framework is that it allows the model to learn a policy that optimally trades off between semantic fidelity (imitation) and physical feasibility (rewards), rather than being forced to fit imperfect data during the initial SFT phase. Typically, after pre-training, supervised fine-tuning should be conducted before RL fine-tuning. However, in the context of controlling real humanoid robots, curating a new dataset of smooth, stable, and consistent motions specifically for the robot is prohibitively expensive. Instead of discarding the valuable semantic information from large-scale human motion datasets, we propose to leverage them as a weak prior. To achieve this, we transform the SFT loss into SFT reward and integrate it directly into the RL fine-tuning process. This unified approach allows the model to simultaneously imitate human motion and optimize for robot-specific physical constraints.

**d) SFT Reward.** While the pre-trained model captures general motion patterns, it lacks task-specific fine-tuning. To address this, we incorporate a supervised fine-tuning (SFT) reward during reinforcement learning. The SFT reward measures the mean squared error (MSE) between the generated keypoints $\hat{\mathcal{P}}_t$ and the ground truth keypoints $\mathcal{P}_t$:

$$\mathcal{R}_{\text{SFT}} = -\frac{1}{T} \sum_{t=1}^{T} \|\hat{\mathcal{P}}_t - \mathcal{P}_t\|_2^2. \tag{9}$$

Then the total reward combines these four components:

$$\mathcal{R}_{\text{total}} = (\mathcal{R}_{\text{smooth}} + \mathcal{R}_{\text{skeleton}} + \mathcal{R}_{\text{consistency}}) + w_{\text{SFT}} \mathcal{R}_{\text{SFT}}, \tag{10}$$

where $w_{\text{SFT}}$ is the hyperparameter balancing the task-specific rewards $(\mathcal{R}_{\text{smooth}}, \mathcal{R}_{\text{skeleton}}, \mathcal{R}_{\text{consistency}})$ and the SFT reward $\mathcal{R}_{\text{SFT}}$. Besides, we can also replace $\mathcal{R}_{\text{smooth}}$ with the normalized $\mathcal{R}_{\text{smooth}}^+$.

When fine-tuning the motion model with RL, we choose the HumanoidVLA language module (Ding et al., 2025) as our base model. The training algorithm is Group Relative Policy Optimization (GRPO) (Shao et al., 2024; Guo et al., 2025), with reference to the open-source implementation TRL-Transformer Reinforcement Learning (von Werra et al., 2020).

## 4 EXPERIMENTS

We conduct a comprehensive evaluation of Humanoid-R0 across five perspectives. First, we evaluate the performance of Humanoid-R0 on the AMASS test set, examining both traditional motion metrics and those relevant to humanoid robot control. Second, we perform systematic ablation study of different SFT weight hyperparameter and compare two types of smooth rewards. Third, we analyze the potential reward hacking during the RL finetuning process and discuss how to avoid this phenomenon. Fourth, we demonstrate the impact of the consistency reward on continuous instruction following tasks within a simulated environment. Last but not the least, we deploy Humanoid-R0 on the real robot G1 and showcase its motion generation capabilities under various continuous commands.

### 4.1 PERFORMANCE COMPARISON

**Baselines** We compare Humanoid-R0 with four baselines:

Table 1: Performance comparison across multiple dimensions including motion quality, text-motion alignment, and physical feasibility. Metrics with ↑ are better when higher, those with ↓ are better when lower. **Bold** denotes the best performance and underlined denotes the second-best.

| Methods | RPrecision↑ | | | FID↓ | MMDist↓ | Diversity | MModality | Skeleton↑ | Smooth↓ |
|---|---|---|---|---|---|---|---|---|---|
| | Top1 | Top2 | Top3 | | | | | | |
| MDM | $0.036^{\pm.003}$ | $0.072^{\pm.006}$ | $0.106^{\pm.007}$ | $0.991^{\pm.066}$ | $8.152^{\pm.029}$ | $3.811^{\pm.098}$ | $3.742^{\pm.201}$ | $0.815^{\pm.135}$ | $\underline{0.107}^{\pm.098}$ |
| T2M-GPT | $\mathbf{0.106}^{\pm.007}$ | $\mathbf{0.195}^{\pm.008}$ | $\mathbf{0.276}^{\pm.008}$ | $0.435^{\pm.033}$ | $\mathbf{7.343}^{\pm.032}$ | $4.572^{\pm.130}$ | $4.562^{\pm.240}$ | $0.770^{\pm.122}$ | $0.111^{\pm.084}$ |
| MotionGPT | $0.036^{\pm.005}$ | $0.072^{\pm.005}$ | $0.104^{\pm.007}$ | $0.250^{\pm.065}$ | $8.358^{\pm.040}$ | $4.446^{\pm.143}$ | $4.334^{\pm.247}$ | $0.748^{\pm.119}$ | $0.120^{\pm.086}$ |
| HumanoidVLA | $0.095^{\pm.007}$ | $0.188^{\pm.006}$ | $0.267^{\pm.009}$ | $\underline{0.170}^{\pm.039}$ | $\underline{7.345}^{\pm.033}$ | $4.526^{\pm.082}$ | $4.226^{\pm.327}$ | $\underline{0.929}^{\pm.072}$ | $0.116^{\pm.094}$ |
| Humanoid-R0 | $\underline{0.102}^{\pm.006}$ | $\underline{0.194}^{\pm.009}$ | $\underline{0.270}^{\pm.006}$ | $\mathbf{0.165}^{\pm.018}$ | $7.370^{\pm.031}$ | $4.262^{\pm.129}$ | $3.909^{\pm.434}$ | $\mathbf{0.938}^{\pm.065}$ | $\mathbf{0.100}^{\pm.081}$ |

- **MDM** (Tevet et al., 2022): A diffusion-based motion generation model that employs a classifier-free approach to create realistic and varied human motions.

- **T2M-GPT** (Zhang et al., 2023a): A text-to-motion generation framework that integrates a VQ-VAE Van Den Oord et al. (2017) for motion representation with a transformer for autoregression.

- **MotionGPT** (Jiang et al., 2023): A unified motion-language model that represents 3D human motions as discrete tokens, enabling joint language modeling for text-to-motion generation.

- **HumanoidVLA** (Ding et al., 2025): A vision-language-action framework that integrates egocentric perception and motion control for autonomous humanoid interaction. Our work builds upon the text-to-motion generation pipeline of HumanoidVLA by focusing on enhancing the physical plausibility of the generated motions for downstream control.

It is important to note that MDM, T2M-GPT, and MotionGPT are pure motion generation models, while our method and HumanoidVLA represent end-to-end control pipelines. Therefore, our evaluation strategy is two-fold: Table 1 compares the motion generation quality of Humanoid-R0 against the baselines using standard metrics. In contrast, Section 4.4 evaluates the deployment performance of the full control pipeline built upon Humanoid-R0 versus the one built upon the base HumanoidVLA model, demonstrating the practical impact of our fine-tuning on continuous task execution.

**Metrics** The evaluation metrics comprehensively assess motion generation performance across four key aspects: (1) Motion Quality is measured by **FID** (Frechet Inception Distance), which evaluates feature similarity between generated and real motions (Gower, 1975) (lower is better); (2) Generation Diversity is assessed using **Diversity** and **MModality** (MultiModality) (Guo et al., 2022), reflecting the variability of generated motions and the ability to generate diverse motions for a given text, respectively; (3) Condition Matching is evaluated via **RPrecision** (Top1/Top2/Top3) and **MMDist** (Multi-modal Distance), measuring text-motion alignment through retrieval accuracy and feature-space distance (Guo et al., 2022) (higher RPrecision and lower MMDist indicate better matching); (4) Physical Plausibility includes **Skeleton** and **Smooth**, where Skeleton ensures anatomical feasibility based on body segment lengths, and Smooth penalizes abrupt transitions to promote stable execution.

**Results** As shown in Table 1, our proposed Humanoid-R0 demonstrates significant advantages over four baseline models in both motion generation quality and physical deployability. It achieves the best results on ***the most important FID*** metrics and skeleton and smoothness metrics closely related to physical deployment, indicating generated motions are not only natural and smooth but also well-suited for real-world execution, offering improved stability and deployability. In terms of text-motion alignment, Humanoid-R0 ranks second across RPrecision and performs competitively on MMDist, both of which are very close to optimal performance, demonstrating strong semantic consistency. While Diversity and MModality show slight drops, this is likely due to the physical constraints introduced during fine-tuning that filters out unrealistic or unstable motions. Notably, T2M-GPT excels in semantic alignment but falls short in physical feasibility, as reflected in its poor Skeleton and FID scores. Overall, the experimental results collectively prove that our Humanoid-R0 optimally balances semantic fidelity with physical feasibility, highlighting its core advantage in enabling reliable and physically feasible control for real-world humanoid robots.

## 4.2 ABLATION STUDY

In the reward function, the weight of the SFT reward is adjusted, while the weights of the other components are fixed at 1. We conduct a detailed ablation study to evaluate the impact of different SFT reward weights on model performance. Simultaneously, we compare the performance of using normalized v.s. unnormalized smooth rewards. Here, we focus on key

metrics of motion quality FID score, the MMDist metric for language-motion alignment, the smoothness and skeletal plausibility of the generated motions. The results have been summarized in Table 2. Results below the base model are colored gray, while the best results are highlighted in **bold**.

As shown in Table 2, both excessively large and small SFT weights degrade FID scores, indicating reduced motion generation quality, regardless of the

Table 2: Performance comparison with different SFT weight on smooth reward and normalized smooth reward.

| Model | FID↓ | MMDist↓ | Skeleton↑ | Smooth↓ |
|---|---|---|---|---|
| Base | 0.1704 | 7.3448 | 0.9290 | 0.1155 |
| Smooth-0.3 | 0.2330 | 7.3573 | **0.9450** | **0.0958** |
| Smooth-0.5 | 0.1648 | 7.3697 | 0.9348 | 0.1005 |
| Smooth-0.8 | **0.1644** | 7.3995 | 0.9358 | 0.1039 |
| Smooth-1.0 | 0.1863 | 7.4072 | 0.9352 | 0.1046 |
| Normalized-Smooth-0.25 | 0.2542 | 7.3452 | 0.9398 | 0.0995 |
| Normalized-Smooth-0.5 | 0.1789 | 7.3452 | 0.9317 | 0.1061 |
| Normalized-Smooth-0.8 | 0.1685 | **7.3217** | 0.9350 | 0.1070 |
| Normalized-Smooth-1.25 | 0.1660 | 7.3677 | 0.9323 | 0.1072 |
| Normalized-Smooth-1.5 | 0.1832 | 7.3347 | 0.9317 | 0.1110 |

smooth reward type. A weight of 0.8 emerges as a relatively balanced choice. The general increase in MMDist indicates that RL finetuning somewhat affects the alignment between motions and text. But the overall performance remains superior compared to baselines. With respect to smoothness and skeletal plausibility, since the objective of RL finetuning is to maximize these two metrics, the finetuned models naturally outperform base models.

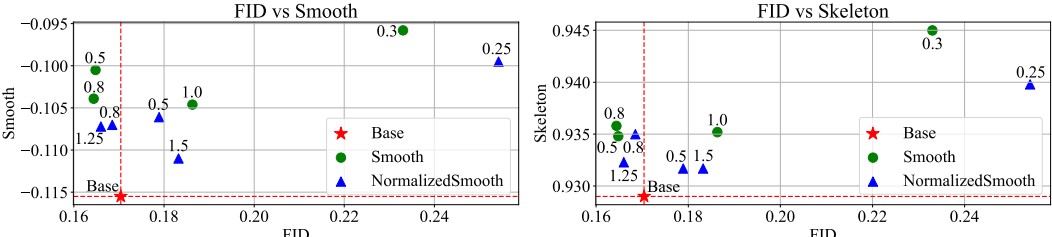

(a) Relationship between FID and smoothness.  (b) Relationship between FID and skeletal plausibility.

To better analyze the impact of smooth reward normalization on performance, we have plotted the relationships between FID and smoothness in Figure 4a, as well as FID and skeletal plausibility in Figure 4b. We expect an RL model that not only maximizes the reward but also generates more precise motions (i.e., lower FID). Therefore, in both plots, models closer to the upper left corner are considered better. It can be observed that, in terms of both smoothness and skeletal plausibility, the unnormalized smooth reward performs better, as indicated by the green dots generally being above the blue triangles. However, when it comes to language-motion alignment, measured by MMDist in Table 2, the normalized smooth reward shows superior performance compared to unnormalized one.

## 4.3 REWARD HACKING

It can be observed in Figure 5 that all rewards initially rise gently during the training process before surging rapidly. This abrupt increase in rewards may indicate that the model has discovered a shortcut to obtaining rewards, which is commonly referred to as reward hacking. Correspondingly, at these steps, both the KL divergence and the length of the generated motion undergo sharp changes, suggesting that the generated motion has significantly deviated from the base model. This essentially

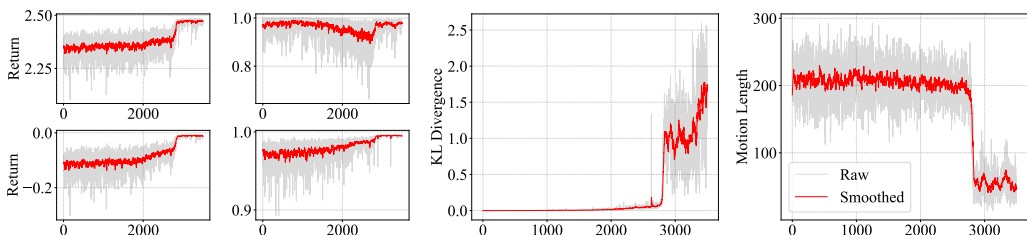

Figure 5: The four smaller images on the left depict the returns during the RL training process, representing total reward, SFT reward, smooth reward, and skeleton reward, respectively. The middle illustrates the changes in KL divergence with respect to the base model during training. The right figure shows the variations in the length of the output motion. All curves have been smoothed.

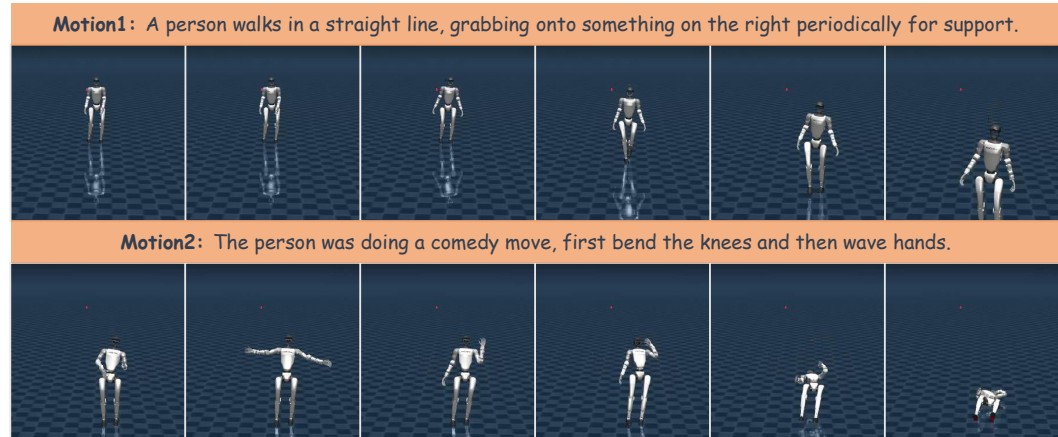

Figure 6: **Base model performance.** The robot is required to complete Motion 2 immediately after finishing Motion 1. The base model successfully completes Motion 1 but fails during Motion 2. This is because it does not account for motion consistency and ignores the state at the end of Motion 1, leading to abrupt motion changes when performing "bend the knees". As a result, the robot loses balance and falls, completing only "wave hands".

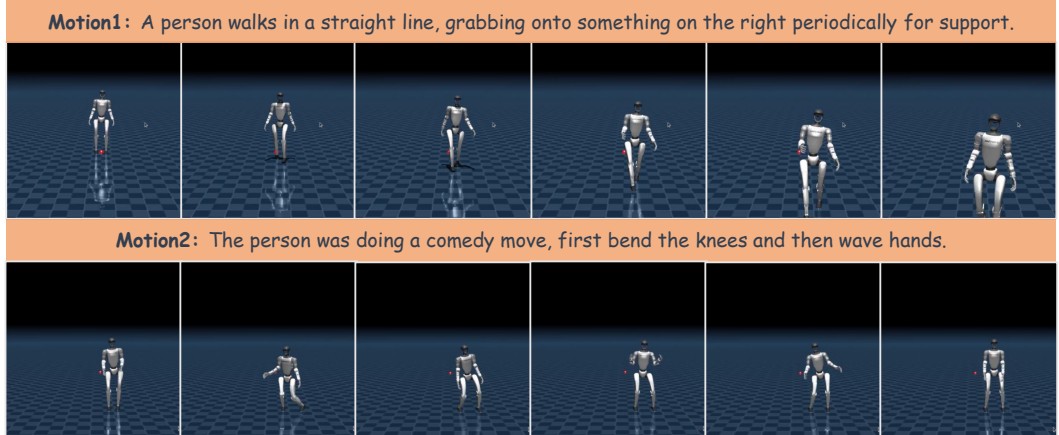

Figure 7: **Humanoid-R0 performance.** Our method ensures smooth transitions between motions and penalizes abrupt changes through the proposed consistency reward (Eq. 8), enabling the successful and seamless execution of both motions.

confirms that reward hacking occurs in the later stages of training. When selecting model checkpoints, we opt for those before the mutation to avoid the potential risk of reward hacking.

## 4.4 SIMULATION VISUALIZATION

We also compare the performance of the original HumanoidVLA pipeline (using its unmodified motion generator) against our proposed pipeline (using the Humanoid-R0 motion generator). As previously analyzed, the robot's current posture must be considered when controlling it; otherwise, the robot may execute abrupt motions with severe consequences. The consistency reward in Humanoid-R0 is specifically designed to address this issue.

In the simulated environment, we tasked the model with continuously generating motions corresponding to two action commands. When executing the first action, the robot started from a standard initial posture and smoothly completed the forward walk. For the second action, which involved squatting and waving, the base model only knew how to perform the action from the standard initial posture, neglecting the robot's left-leaning posture at that moment. As a result, the robot fell backward when squatting. In contrast, the RL-fine-tuned model, which fully considered the robot's current posture, smoothly transitioned and successfully completed the squatting and waving action.

Figure 8: Deployment on a humanoid robot G1. The robot successfully performs consecutive diverse motions (e.g., waving, walking, squatting) with smooth and stable motions.

### 4.5 REAL WORLD ROBOT DEPLOYMENT

Our framework demonstrates robust real-world deployment capabilities on physical humanoid robots. As shown in Figure 8 [1], the proposed method successfully executes continuous commands with smooth and stable motions, showcasing its reliability for practical applications. The robot seamlessly transitions between tasks, maintaining balance and stability throughout complex sequences such as waving, walking, and squatting. These results validate our approach's ability to generate physically feasible motions that are deployable on real-world systems.

## 5 CONCLUSION

This paper presents a framework of motion generation model fine-tuning for humanoid robot control. By integrating smoothness, skeletal plausibility, and keyframe consistency into the RL fine-tuning process via GRPO, our method, Humanoid-R0, bridges the gap between language-conditioned motion synthesis and real-world execution. Experimental results demonstrate that our approach significantly improves motion quality and deployability across both simulated and real-world settings. Experimental results demonstrate that our approach significantly improves motion quality and deployability across both simulated and real-world settings, paving the way for more robust and intuitive language-driven control of humanoid robots.

---

[1]Redundant background elements are removed for clarity, but robot and ground markings remain unchanged.

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

## A  LLMs Usage Statement

We used large language models (LLMs) solely for the purpose of grammar checking, sentence polishing, and improving the overall readability of the manuscript. The use of LLMs was strictly limited to linguistic refinement and did not involve any aspect of the core research methodology. All technical contributions, ideas, analyses, and conclusions presented in this paper are entirely the work of the authors.

## B  Limitations

Our framework demonstrates strong performance in generating semantically aligned and physically feasible motions; however, it also has some limitations.

1. Inference Speed: The current inference speed is relatively slow, and the generated motions must be stored before execution, making real-time interaction infeasible. We may need to consider lightweight optimization approaches to overcome these challenges, in order to achieve real-time instruction following and more sensitive humanoid control.

2. Scope of Tasks and Interactions: Our evaluation focuses on fundamental whole-body locomotion and gestures. Extending this framework to complex manipulation tasks involving object interaction remains a significant challenge, primarily due to the lack of large-scale, egocentric visual-language-motion datasets needed to ground actions in the environment.

3. Generalization to Diverse Hardware: The skeletal plausibility reward is tailored to the specific joint configuration of the G1 robot. While the framework is general, adapting it to robots with different kinematic structures would require re-calibration of the reward parameters. However, we envision that a meta-learning or adaptation framework could be developed in the future to automatically tune these parameters for new robotic platforms, significantly enhancing the generalizability of our approach.

## C  Social Impact

Our work contributes to the development of language-conditioned humanoid control systems, which may have broad applications in assistive robotics, home automation, and human-robot collaboration. By enabling intuitive command-based control, it can lower the barrier for non-expert users to interact with physical robots in daily environments. Additionally, our approach significantly improves the deployment success rate of existing humanoid robot models, making them more reliable and practical for real-world use cases.

## D  Hyperparameters of Reward Functions

For RL fine-tuning of the motion generation model, the design of the reward function is highly critical. We summarize below the meanings of the hyperparameters included in each reward function, as well as their specific values.

**Smoothness**  The non-normalized smooth reward $\mathcal{R}_{\text{smooth}}$ does not contain any additional hyperparameters, whereas the normalized reward $\mathcal{R}^{+}_{\text{smooth}}$ includes pre-defined mean $\mu$ and std $\sigma$. During the maximization process of the normalized smooth reward, the smooth reward will approach $\mu$. Selecting an appropriate $\mu$ can stably achieve smoothness and prevent reward hacking that leads to motionless actions.

Table 3: Hyperparameters of Smooth Reward.

| | |
|---|---|
| $\mu$ | **0.08** (Hyperparameters) |
| $\sigma$ | **0.104177** (also the std of AMASS datasets) |

Table 4: Hyperparameters of Skeleton Reward.

| Skeletons | Ranges |
|---|---|
| Elbow-Hand (Joints $10 \sim 11$ and $13 \sim 14$) | $[0.26, 0.27]$ |
| Toe-Ankle (Joints $3 \sim 4$ and $7 \sim 8$) | $[0.14, 0.15]$ |
| Knee-Ankle (Joints $2 \sim 3$ and $6 \sim 7$) | $[0.42, 0.44]$ |

**Skeletal Plausibility**    We aim to ensure that the output motion matches the real G1 robot in terms of certain key skeletal lengths. To achieve this, I have defined the length ranges for the following six joints. When the skeletal lengths fall within these specified ranges, the skeleton reward is set to 1. Otherwise, it will decay exponentially.

# E    DEPLOYMENT DETAILS

The video presented in Section 4.5 of the main text had a cluttered background, so we removed some of the background elements during the presentation. It is important to note that the humanoid robot itself was not subjected to any manual modifications.

In contrast, the video provided in the appendix has been improved. To make the robot's hand movements more visible, we added a silicone hand. We also changed the location to achieve a cleaner background. It should be noted that the person holding the rope is **purely for safety reasons** and does not affect the robot's motion. Additionally, the basic video corresponds to the sequence of motions demonstrated in Section 4.5. We also randomly rearranged these seven motions in different orders.

