# OpenReview forum: "Humanoid-R0: Bridging Text-to-Motion Generation and Physical Deployment via RL"
_ICLR.cc/2026/Conference — ICLR 2026 Conference Withdrawn Submission_

### Official Review · Reviewer_LHZM · 2025-10-31

**Soundness:** 1
**Presentation:** 2
**Contribution:** 1
**Rating:** 2
**Confidence:** 5

**Summary:**

The paper proposes Humanoid-R0, a reinforcement-learning fine-tuning approach that adapts an existing text-to-motion generator (HumanoidVLA) for safe execution on a physical humanoid (G1). Three physically motivated rewards—smoothness, skeletal plausibility, and key-frame consistency—are formulated, and the authors fold the usual supervised-fine-tuning loss into the RL objective via an “SFT reward”. Using GRPO for policy updates, the fine-tuned generator produces motions that, once tracked by a low-level PD controller, are claimed to execute more stably than the base model. Quantitative results on AMASS and qualitative demonstrations in simulation and on a real robot are reported.

**Strengths:**

* **End-to-end deployment:** The paper takes the commendable step of validating on real hardware (Figure 8), something still rare in text-to-motion work.
* Table 1 indicates that the fine-tuned model matches or slightly improves the base model in FID (0.170 → 0.165) while improving the smoothness metric (0.116 → 0.100) and skeleton score (0.929 → 0.938). *(Note: The paper reports smoothness as 0.116 -> 0.100, where lower is better, so this is an improvement).*
* **Ablation study** (Table 2, Figures 4a–4b) explores SFT-weight choices and the effect of reward normalisation, giving some insight into trade-offs.

**Weaknesses:**

* **Missing comparisons** to recent RL-based text-to-motion fine-tuning approaches that aim to improve physical feasibility, such as RobotMDM [1] and RLPF [2], which address similar objectives. This greatly weakens the novelty claims.
* **Ad-hoc reward design:** The specific rewards are overly ad-hoc (hand-crafted) and unlikely to cover all necessary physical scenarios. Is physical feasibility truly captured by only these three metrics? This seems far from sufficient. I believe this approach is, **in principle**, far less sound than the methods in RobotMDM [1] or RLPF [2].
* **Concerns about generation quality:** The reported R-Precision (Top 1) of **~0.1** seems **unusually low**, especially when paired with a strong (low) FID score. Typically, in text-to-motion literature, these metrics are inversely correlated: methods using TMR-based retrieval (e.g., MotionStreamer[3]) achieve high R-Precision (~0.7) but larger FIDs, while methods using feature-space distance (Guo et al., 2022) achieve lower R-Precision (~0.5) but very low FIDs. The results presented in this paper (R-Precision ~0.1) do not align well with either established pattern, raising doubts about the generation quality, specifically regarding **semantic fidelity**. More video visualizations of the generated motions are needed to allay these concerns.
* **Limited real-robot evaluation:** Only a single 7-command demo on one robot is shown. No success-rate statistics, MPJPE, or MPKPE are provided. Without quantitative metrics, it is hard to gauge robustness. Furthermore, the lack of tests in multiple simulators makes it difficult to judge the generalizability of the claimed physical feasibility.
* **Baseline disparity:** Three baselines (MDM, T2M-GPT, MotionGPT) are pure motion generators; moreover, they are all relatively **dated baselines**. Humanoid-R0 relies on a powerful controller underneath. The fairness of comparing surface keypoints alone without simulating tracking controllers is questionable.
* **Marginal quantitative gains:** Improvements in Table 1 over HumanoidVLA are modest (<3 % on FID, <0.6 % on skeleton, negligible on R-Precision). Confidence intervals overlap for several metrics, so statistical significance is unclear.
* **Methodological clarity:**
    * Equation (10) sums rewards of different scales without normalisation; no gradient-scale balancing or adaptive weighting is discussed.
    * The consistency reward (Eq. 8) is a distance penalty (a loss) but appears to be added positively in the total reward (Eq. 10); the signs seem inconsistent.
    * It is unclear whether GRPO updates only the generator or also the language encoder; Section 3.3 hints at the latter, but Figure 1 colors only the generator.
* **Reward hacking not solved:** Authors admit (Figure 5) that training diverges; they manually pick an early checkpoint instead of addressing the root cause, indicating instability. This strongly suggests issues with the **reward design or algorithmic implementation**.
* **Computational cost** and real-time constraints (acknowledged in Limitations) are not reported—how many RL steps, GPU hours, or what is the real-time factor on the robot?
* **Clarity issues:** Figure 2 joints are indexed but not mapped to robot DoFs; footnotes in Section 4.5 are oddly formatted; the term “Humanoid-R0” sometimes refers to the generator only, other times to the full pipeline.

**Questions:**

1.  How were $\mu = 0.08$ and $\sigma = 0.104$ in **Eq. (6)** chosen beyond “same as AMASS std”? Did you test sensitivity to these values?
2.  In **Eq. (10)**, the consistency term **(Eq. 8)** appears to be added, but Eq. (8) is defined as a distance (a loss) to be minimised. Should the sign be negative?
3.  Does GRPO update the language encoder or only the token generator? If the former, how do you prevent catastrophic forgetting and maintain text alignment stability?
4.  The paper lacks a **detailed explanation** of the Humanoid Robot Controller. What are the specific inputs to the control policy (Eq. 4)? Is it a Teacher-Student architecture (e.g., similar to ASAP)?
5.  Regarding the **SFT Reward (Eq. 9)**: How is the MSE loss calculated when the generated motion sequence $\hat{P}$ and the ground-truth motion $P$ have different lengths?
6.  How does the paper formally define "Physical Feasibility" and "Semantic Fidelity"? The current definitions seem implicit and are only partially covered by the chosen metrics.


### Potentially Missing Related Work
[1] Serifi A, Grandia R, Knoop E, et al. Robot motion diffusion model: Motion generation for robotic characters[C]//SIGGRAPH asia 2024 conference papers. 2024: 1-9.
[2] Yue J, Wang Z, Wang Y, et al. RL from Physical Feedback: Aligning Large Motion Models with Humanoid Control[J]. arXiv preprint arXiv:2506.12769, 2025.
[3] Xiao L, Lu S, Pi H, et al. MotionStreamer: Streaming Motion Generation via Diffusion-based Autoregressive Model in Causal Latent Space[J]. arXiv preprint arXiv:2503.15451, 2025.

---

### Official Review · Reviewer_k8mz · 2025-10-31

**Soundness:** 2
**Presentation:** 2
**Contribution:** 2
**Rating:** 2
**Confidence:** 5

**Summary:**

This paper presents HUMANOID-R0, a text-to-motion generation framework fine-tuned with reinforcement learning (RL) to bridge the gap between simulated motion synthesis and real-world deployment on humanoid robots. By quantifying and formulating key physical requirements—smoothness, skeletal plausibility, and keyframe consistency—as explicit reward functions, and integrating a supervised fine-tuning (SFT) reward, the method aims to produce motions that are not only semantically aligned with text instructions but also robustly deployable on actual robots. HUMANOID-R0 extends the motion generation component of HumanoidVLA, uses Group Relative Policy Optimization (GRPO) for RL-based refinement, and demonstrates notable improvements across both simulation and real-robot deployment, as supported by quantitative benchmarks and qualitative results.

**Strengths:**

- The work directly addresses the pressing challenge of deploying text-based motion generation models on physical humanoid robots.
- The paper proposes well-motivated rewards for smoothness, skeletal plausibility, and keyframe consistency, providing mathematical definitions.
- The deployment and long-horizon sequence execution on the G1 robot demonstrate convincing real-world utility.

**Weaknesses:**

- **Incomplete Discussion of Closely Related RL-Augmented Text-to-Motion Methods**: The paper lacks a systematic discussion and empirical comparison with several representative “text-to-motion + physics/preference RL” approaches, including **RobotMDM** (model-driven, robot-deployable motion generation), **ReinDiffuse** (RL refinement of diffusion models to suppress physical artifacts), **RLPF** (preference/reward-learning–based motion optimization), **Motion-R1** (instruction-aligned large motion model paradigm), **MotionRL** (multi-objective rewards balancing text fidelity and human biomechanics), and **Morph** (RL-regularized motion prior for physically plausible, morphology-aware control). We recommend adding a subsection that, for each method, clarifies (i) reward design (text alignment, physical feasibility, human preference), (ii) scope (pure kinematic trajectories vs. policy rollouts with contact dynamics), (iii) training strategy (RLHF/offline RL/online RL, stabilization and anti–reward hacking measures), (iv) evaluation axes (contact/slip/penetration rates, joint-limit violations, long-horizon stability, zero-shot generalization), and (v) real-robot validation and deployment cost. Provide a one-to-one comparison table against HUMANOID-R0—highlighting your GRPO, smoothness/skeletal-plausibility/keyframe-consistency rewards, and SFT-as-intrinsic reward—and include head-to-head results under the same data and control stack to more clearly position your contributions and advantages.

- **Insufficient Novelty in Reward Design**: While the operationalization of smoothness, skeletal plausibility, and keyframe consistency as rewards is a reasonable engineering step, the core mathematical forms (e.g., $L_2$ norm smoothness, distance-based skeleton consistency, mean squared error SFT reward) are standard choices. The most unique aspect—the normalized smooth reward—does not represent a major advance over prior work in reward normalization or anti-hacking strategies.
- **Insufficient Analysis of the Controller and Deployment Stack.** The paper under-specifies the control pipeline and its learning/transfer details. Please expand on: (i) **Goal & observation modeling**—precisely define command/goal parameterization (task frames, waypoints, keyframes, language-conditioned goals), observation vectors (proprioception, estimated contacts, IMU, joint/Cartesian states), history windows, rates, delays, and normalization; (ii) **Controller design**—action space (joint torques vs. target positions/velocities/impedances), control frequency, latency compensation, safety limits (torque/current/velocity clamps), contact handling, and how the policy interfaces with low-level PD/MPC/whole-body QP; (iii) **Controller reward**—all terms with formulas, weights, schedules, and curriculum (tracking error, contact consistency, energy/torque penalties, foot slip/impact costs), plus anti–reward-hacking strategies (normalization, clipped advantages, terminal penalties); (iv) **Simulation details**—physics engine, integrator/solver settings, contact/friction models, actuator and sensor models, time-step, system identification, and domain randomization ranges (mass/inertia, friction, compliance, latency, sensor noise, terrain); (v) **Sim-to-real transfer**—calibration/SysID procedure, latency and noise injection, actuator nonlinearity modeling, residual/adaptation policies, safety monitors, and any on-robot fine-tuning; (vi) **Controller experiments**—ablate observation subsets, action parametrizations, reward weights, and control rates; report stability margins, tracking RMSE, peak/average torques and power, contact slip/impact metrics, controller saturation, recovery counts, success under perturbations (pushes, payloads, friction changes, terrain), and long-horizon reliability with confidence intervals. A dedicated “Controller & Sim-to-Real” section with a comparison table (policy interface × sim config × transfer strategy) and controlled head-to-head trials would substantively strengthen the paper’s technical rigor and reproducibility.

- **Insufficient Visual Evidence (simulation ↔ real-robot)**. The paper provides too few verifiable videos to substantiate deployability, robustness, and the claimed benefits of the smoothness design. Include paired, time-synchronized clips for each task—side-by-side sim vs. real under identical seeds/commands, fixed camera views, and unbroken takes—and release matching raw logs and replayable trajectories. Critically, add ablation videos comparing (a) without vs. (b) with the **smoothness reward** under the same conditions, with overlays for joint velocity/acceleration/jerk, contact phases, foot-slip traces, and tracking error. Without these side-by-side and ablation visualizations, the smoothness contribution and sim-to-real fidelity remain difficult to verify.

**Questions:**

1. Can the authors explicitly clarify how the normalized smooth reward ($\mathcal{R}_{\text{smooth}}^{+}$) is parametrized and how hyperparameters $\mu$, $\sigma$, and unspecified operands like $e$ are selected and influence reward scale, especially in Equation (before Table 3 in Appendix D)? A clear description here is critical for reproducibility.
2. Could you provide a more detailed quantitative analysis of real-world deployment failures (e.g., falls, actuator saturation), safety metrics, or repeated trial robustness, beyond the qualitative illustrations (Figure 8)?
3. Could you comprehensively document the controller and sim-to-real pipeline, including: (a) goal/observation parameterization and normalization; (b) action space (torque/position/impedance), control frequency, and latency compensation; (c) controller-reward terms with formulas, weights/schedules, and anti–reward-hacking measures; (d) simulator settings (engine, integrator, contact/friction models, actuator/sensor models, timestep) and domain-randomization ranges; (e) SysID, latency/noise injection, and hardware safety limits; and (f) ablations/head-to-head experiments isolating each design choice? Providing config files (e.g., YAMLs) would substantially improve reproducibility.

---

### Official Review · Reviewer_VQdE · 2025-11-01

**Soundness:** 2
**Presentation:** 3
**Contribution:** 2
**Rating:** 2
**Confidence:** 4

**Summary:**

This paper fine-tunes a text-to-motion model with RL to make its generated motions safer and more stable for real humanoids. It introduces three motion-based reward terms—smoothness, skeletal consistency, and keyframe consistency—and combines them with a transformed SFT loss to shape policies that are both physically feasible and semantically aligned. Experiments include standard motion metrics, ablations, simulation rollouts, and real deployment on a Unitree G1 performing sequential natural-language commands. Visual (images only) results showed a few text to motion examples in simulation and the real world.

**Strengths:**

- Rewarding physical plausibility and initial pose continuity directly targets an important failure mode for text-driven robot motion.
- The KL/motion-length plots show awareness of RL failure modes and checkpoint selection to avoid degeneration.

**Weaknesses:**

- The qualitative evaluation is very limited. Figure 6 and Figure 7 only illustrate a single two-step motion sequence, and do not cover the breadth of actions the model claims to support. Similarly, Figure 8 is difficult to interpret visually — the static snapshots do not convey motion smoothness, stability, or transition quality. For a paper focused on motion quality and deployability, higher-quality motion visualization is essential. Ideally, a diverse set of motion sequences should be shown side-by-side with baselines and ablations.
    - Real-world deployment videos are mentioned but not provided.
    - Real-world results lack quantitative numbers.
- Tracking performance on the AMASS test set is not found. (mentioned in L260).

**Questions:**

Access to the real-world depoyment videos would be appreciated.

---

### Official Review · Reviewer_t3cP · 2025-11-01

**Soundness:** 3
**Presentation:** 2
**Contribution:** 2
**Rating:** 2
**Confidence:** 3

**Summary:**

This paper proposes a framework that finetunes a pre-trained text-to-motion model using reinforcement learning (RL) with a set of rewards  to encourage the transferability of such generated motions to physical hardware. The proposed rewards are based on a smoothness metric, skeletal palusibility metric, and keyframe consistency metric proposed by the authors. The proposed method is test on AMASS dataset to show it beats state-of-the-art models on the text-to-motion problem. In another set of experiments, authors demonstrate their contribution in simulation as well as a humanoid robot.

**Strengths:**

This work addresses an important and timely problem. Current text-to-motion models can generate high-quality kinematic motions consistent with text prompts. However, these motions are often not directly transferable to physical hardware due to issues such as skeletal physibility and smoothness issues. Developing algorithms that produce robust, high-quality motions that can be readily transferred to physical robots is therefore a valuable research direction.

**Weaknesses:**

* My biggest concern of this work is the lack of results supporting the contribution claims. Proposed method is supposed to yield a text-to-action model, wiht HumanoidVLA as the text-to-motion component of it - which finetuned with RL such that its outputs are better transfered the the controller. In Table 1, effectiveness of HumanoidVLA is demonstrated as a text-to-motion model. However, it is hard to see how effective the model is as a text-to-action model as the provided simulation / robot results are compressed into a small number of images. Line 073 refers us to the video included in the supplementary material but it looks like the supplementray material is not uploaded. In the absence of quantitative evaluation for this task, it is necessary to provide video results to see how effective the proposed method is.

* I am curious how this method compares with recent motion tracking and humanoid control works such as BeyondMimic. BeyondMimic focuses on transferring kinematic motions to real hardware. Given that modern T2M models are already capable of generating high-quality kinematic motions, a robust motion tracker might suffice to bridge text to physical motion. Considering the higher quality of motions generated by these recent methods, the novelty and contribution of this work appear limited.

**Questions:**

* The paper repeatedly refers to “large motion models.” Is this an established term?

* Line 135: Is the text-to-motion model first pretrained on text-to-video data? If yes, what dataset? Is it then finetuned on motion data? If yes, what dataset is used exactly?

* Table 1 seems to show the results of Humanoid-R0 and other text-to-motion models on the AMASS dataset. I think the AMASS dataset is not annotated with text prompts. If that's the case, where are the RPrecision metrics computed?

* Why is the output of the text-to-motion model referred to as keyframes? For example, in line 131, it is saied that $\pi_{\texttt{Text2Motion}}$ generates the keypoint trajectory.

* Since the HumanoidVLA model is refered to as the base model, it would be useful to include a description of this method and what it does.

---

### Note · Authors · 2025-11-12

I have read and agree with the venue's withdrawal policy on behalf of myself and my co-authors.